# Diagnostic Role and Prognostic Impact of PSAP Immunohistochemistry: A Tissue Microarray Study on 31,358 Cancer Tissues

**DOI:** 10.3390/diagnostics13203242

**Published:** 2023-10-18

**Authors:** Laura Sophie Tribian, Maximilian Lennartz, Doris Höflmayer, Noémi de Wispelaere, Sebastian Dwertmann Rico, Clara von Bargen, Simon Kind, Viktor Reiswich, Florian Viehweger, Florian Lutz, Veit Bertram, Christoph Fraune, Natalia Gorbokon, Sören Weidemann, Claudia Hube-Magg, Anne Menz, Ria Uhlig, Till Krech, Andrea Hinsch, Eike Burandt, Guido Sauter, Ronald Simon, Martina Kluth, Stefan Steurer, Andreas H. Marx, Patrick Lebok, David Dum, Sarah Minner, Frank Jacobsen, Till S. Clauditz, Christian Bernreuther

**Affiliations:** 1Institute of Pathology, University Medical Center Hamburg-Eppendorf, 20251 Hamburg, Germany; l.tribian@uke.de (L.S.T.); m.lennartz@uke.de (M.L.); d.hoeflmayer@uke.de (D.H.); s.dwertmann-rico@uke.de (S.D.R.); c.von-bargen@uke.de (C.v.B.); s.kind@uke.de (S.K.); v.reiswich@uke.de (V.R.); f.viehweger@uke.de (F.V.); f.lutz@uke.de (F.L.); v.bertram@uke.de (V.B.); c.fraune@uke.de (C.F.); n.gorbokon@uke.de (N.G.); s.weidemann@uke.de (S.W.); c.hube@uke.de (C.H.-M.); a.menz@uke.de (A.M.); r.uhlig@uke.de (R.U.); t.krech@uke.de (T.K.); a.hinsch@uke.de (A.H.); e.burandt@uke.de (E.B.); g.sauter@uke.de (G.S.); m.kluth@uke.de (M.K.); s.steurer@uke.de (S.S.); p.lebok@uke.de (P.L.); d.dum@uke.de (D.D.); s.minner@uke.de (S.M.); f.jacobsen@uke.de (F.J.); t.clauditz@uke.de (T.S.C.); c.bernreuther@uke.de (C.B.); 2Department of General, Visceral, and Thoracic Surgery, University Medical Center Hamburg-Eppendorf, 20251 Hamburg, Germany; n.de-wispelaere@uke.de; 3Institute of Pathology, Clinical Center Osnabrueck, 49076 Osnabrueck, Germany; 4Department of Pathology, Academic Hospital Fuerth, 90766 Fuerth, Germany; andreas.marx@klinikum-fuerth.de

**Keywords:** PSAP, immunohistochemistry, tissue microarray, prognosis, prostate cancer

## Abstract

Prostate-specific acid phosphatase (PSAP) is a marker for prostate cancer. To assess the specificity and prognostic impact of PSAP, 14,137 samples from 127 different tumor (sub)types, 17,747 prostate cancers, and 76 different normal tissue types were analyzed via immunohistochemistry in a tissue microarray format. In normal tissues, PSAP staining was limited to the prostate epithelial cells. In prostate cancers, PSAP was seen in 100% of Gleason 3 + 3, 95.5% of Gleason 4 + 4, 93.8% of recurrent cancer under androgen deprivation therapy, 91.0% of Gleason 5 + 5, and 31.2% of small cell neuroendocrine cancer. In non-prostatic tumors, PSAP immunostaining was only found in 3.2% of pancreatic neuroendocrine tumors and in 0.8% of diffuse-type gastric adenocarcinomas. In prostate cancer, reduced PSAP staining was strongly linked to an advanced pT stage, a high classical and quantitative Gleason score, lymph node metastasis, high pre-operative PSA levels, early PSA recurrence (*p* < 0.0001 each), high androgen receptor expression, and *TMPRSS2:ERG* fusions. A low level of PSAP expression was linked to PSA recurrence independent of pre- and postoperative prognostic markers in ERG-negative cancers. Positive PSAP immunostaining is highly specific for prostate cancer. Reduced PSAP expression is associated with aggressive prostate cancers. These findings make PSAP a candidate marker for prognostic multiparameter panels in ERG-negative prostate cancers.

## 1. Introduction

Prostate-specific acid phosphatase (PSAP, syn. prostatic acid phosphatase, PAP)—is produced in the epithelial cells of the prostate gland. PSAP dephosphorylates macromolecules in acidic conditions (pH 4–6), but its physiological substrates are not fully known [1,2]. PSAP is assumed to directly influence sperm motility and viability [3]. Alternative splicing generates three types of PSAP transcripts, namely, a transmembrane PSAP, a secretory PSAP, and a cellular PSAP. The molecular mechanisms controlling PSAP protein expression are not completely understood. Factors known to be involved in the regulation of PSAP expression include androgen, androgen receptor, NF-κB, TNF-alpha, and IL-1 factors (summarized in [4,5]).

Due to the specificity of its expression to normal prostate epithelium, immunohistochemical PSAP analysis—along with prostate-specific antigen (PSA)—is often used to prove the prostatic origin of metastatic lesions (summarized in [4]). However, PSAP-negative prostate cancers have been found in 5–41% of cases [6,7,8]. Reduced PSAP expression is especially common in poorly differentiated cancers [9,10]. Accordingly, some studies on 54–78 patients have suggested that reduced PSAP levels may be linked to poor patient prognosis [11,12], although other studies on 19–68 patients have not confirmed these observations [13,14]. The utility of PSAP immunohistochemistry (IHC) for the determination of a tumor’s site of origin has been challenged by reports describing PSAP positivity in up to 40% of colorectal cancers [15], 33% of non-small cell lung cancer (NSCLC) samples [16], 30% of ovarian cancers [15], 10% of pancreatic cancers [15], 13% of breast cancers [16], 40% of renal cancers [16], and 68% of carcinoid tumors of the gastrointestinal tract [17]. Technical factors, such as staining protocols and the antibodies used, and the different definitions of thresholds for positivity, as well as possible selection bias concerning the analyzed tumors, may have caused these discrepancies. 

To comprehensively assess the prevalence of PSAP expression in many human tumor types, and to estimate the putative diagnostic value of PSAP IHC in prostate cancer, tissue microarrays (TMAs) made from more than 30,000 tumor samples from 132 different tumor types and subtypes, as well as 76 normal tissue categories, were studied.

## 2. Materials and Methods

Tissue Microarrays (TMAs). The process of tissue microarray (TMA) manufacturing has been described in detail in previous studies [18,19]. Our normal tissue sample TMA was composed of 8 samples from 8 different donors for each of 76 different normal tissue types (608 samples on one slide). The multi-tumor TMAs contained a total of 13,611 extra-prostatic primary tumors from 127 different tumor types and subtypes. The composition of both normal and multi-tumor TMAs is described in detail in the Results section. In addition, a prognosis TMA built from prostate cancer samples from radical prostatectomy specimens, derived from 17,747 patients [20] who underwent surgery between 1992 and 2014 at the Department of Urology and the Martini Clinic at the University Medical Center Hamburg-Eppendorf, was analyzed. Follow-up data were available from 14,464 (81.5%) of these patients, with a median follow-up period of 48 months. The histopathological and clinical data are summarized in Appendix A. The molecular database attached to the prostate cancer TMA contained results on androgen receptor (AR) expression (*n* = 7776, expanded from [21]), ERG expression (*n* = 13,089, expanded from [22]), and ERG break-apart fluorescence in situ hybridization (FISH) analysis (*n* = 7225, expanded from [22]). The tissue samples included in this study were retrieved from the archives of the Institutes of Pathology, University Hospital of Hamburg, Germany, the Institute of Pathology, Clinical Center Osnabrueck, Germany, and Department of Pathology, Academic Hospital Fuerth, Germany. All tissue samples were formalin-fixed (4% buffered formalin) and paraffin-embedded. The TMA was manufactured with a tissue spot diameter of 0.6 mm. The usage of anonymized tissue samples for TMA creation and research was approved by local laws (HmbKHG, §12) and by the local ethics committee (Ethics Commission Hamburg, WF-049/09), and the study was carried out in compliance with the Helsinki Declaration. 

Immunohistochemistry (IHC). All experiments were carried out on the same day using freshly cut TMA sections. The sections were immersed in xylol for paraffin removal before alcoholic rehydration. Heat-induced antigen retrieval was performed for 5 min in an autoclave at 121 °C in pH 7.8 Dako Target Retrieval Solution^TM^ (Agilent, CA, USA; #S2367). Dako Peroxidase Blocking Solution^TM^ (Agilent, CA, USA; #52023) was used to inhibit endogenous peroxidase activity. The primary antibody against PSAP protein (mouse recombinant, MSVA-452M, MS Validated Antibodies, Hamburg, Germany, #2521-452M) was incubated at 37 °C for 60 min at a dilution of 1:150. The bound antibody was detected with the EnVision Kit^TM^ (Agilent, CA, USA; #K5007), according to the manufacturer’s directions, before counterstaining the sections with hemalaun. All tissue spots were analyzed by a single experienced pathologist. Several studies have demonstrated that involving multiple pathologists or non-pathologists to read the same TMA slides would not significantly impact the study outcome [23,24,25,26]. For tumor tissues, the percentage of PSAP-positive tumor cells was estimated, and the staining intensity was semi-quantitatively recorded (0, 1+, 2+, 3+). For statistical analyses, the staining results were categorized into four groups, as follows: negative, no staining at all; weak staining, staining intensity of 1+ in ≤ 70% or a staining intensity of 2+ in ≤ 30% of tumor cells; moderate staining, staining intensity of 1+ in > 70%; staining intensity of 2+ in > 30%, but in ≤ 70% or staining intensity of 3+ in ≤ 30% of tumor cells; strong staining, staining intensity of 2+ in > 70% or staining intensity of 3+ in > 30% of tumor cells. For the purposes of antibody validation, the normal tissue TMA was also analyzed for PSAP expression by using the monoclonal mouse anti-PSAP antibody PASE/4LJ (mouse monoclonal, DAKO/Agilent, CA, USA, #M0792, 1:2000, pH high) in the Autostainer Link 48 (Agilent, CA, USA), used according to the manufacturer’s directions.

Statistics. Statistical calculations were performed using the JMP 16.0.0 software (SAS Institute Inc., Cary, NC, USA). Contingency tables and the chi^2^-test were performed to search for associations between PSAP staining, molecular parameters, and tumor phenotype. Survival curves were calculated according to the Kaplan–Meier model. The log-rank test was applied to detect significant survival differences between groups. Cox proportional hazard regression analysis was performed to test the statistical independence and significance between pathological, molecular, and clinical variables. A *p*-value of ≤0.05 was considered to be statistically significant. 

## 3. Results

### 3.1. Technical Issues

Overall, 10,346 (76%) out of 13,611 tumor samples were analyzable in our multi-tumor TMA and 15,455 (87%) out of 17,747 tumor samples were analyzable in our prostate cancer TMA analysis. The rate of interpretable cases was higher for the prostate cancer TMA than for the multi-tumor array (MTA) because the prostate cancer specimens were all taken from prostatectomy specimens, in which the standard block thickness was greater than average. The thickness of the block reflected the length of the tissue cylinders removed during TMA manufacturing and, therefore, the number of sections that could be taken. Reasons for non-interpretable samples included a lack of sufficient amounts of tumor cells or a lack of entire tissue spots in the TMA section. More than three samples of each normal tissue type were evaluable in our normal tissue TMA analysis.

### 3.2. PSAP in Normal Tissues

Strong cytoplasmic PSAP staining was seen in the prostate glandular cells. The PSAP staining was often not restricted to cancer cells but also involved the stroma. This obviously represents a contamination artifact, which is often seen in the case of very highly expressed proteins [27,28]. Weak cytoplasmic PSAP staining was seen in a few tubular cells in the kidney. PSAP immunostaining was absent in all other tissues, including all epithelial cells of the gastrointestinal and the genitourinary tract, fallopian tube, endometrium, endocervix, placenta, gallbladder, liver, pancreas, salivary and bronchial glands, breast glands, Brunner glands, thyroid, pituitary gland, adrenal gland, parathyroid gland, testis, epididymis, seminal vesicle, non-keratinizing and keratinizing squamous epithelium of various different sites, skin appendices, all mesenchymal tissues, hematopoietic and immune cells, and the brain. All cell types found to be PSAP-positive by MSVA-452M were also positive when using PASE/4LJ. Representative images of PSAP-positive and negative tissues are shown for both antibodies in Figure 1.

### 3.3. PSAP in Neoplastic Tissues

PSAP staining was almost exclusively seen in prostate cancers, with PSAP positivity in 96.9% of 15,455 prostate cancers, and only occurred in 2 of the 127 evaluated extra-prostatic tumor categories (diffuse-type gastric adenocarcinoma and neuroendocrine tumors of the pancreas; Table 1, Appendix A). In prostate cancers, the rate of positivity decreased from Gleason 3 + 3 (100%) to Gleason 4 + 4 (95.5%), as with recurrent prostate cancer under androgen deprivation therapy (93.8%), Gleason 5 + 5 = 10 (91.0%), and small cell neuroendocrine (31.2%) prostate cancer. PSAP staining was only seen in 4 (0.03%) out of 13,611 extra-prostatic primary tumors from 2 different tumor categories. PSAP-positive extra-prostatic cancers included 3 of 94 (3.2%) neuroendocrine tumors of the pancreas and 1 of 129 (0.8%) gastric adenocarcinomas (Table 1, Appendix A). Figure 2 shows examples of PSAP-positive tumors.

### 3.4. PSAP, Tumor Phenotype, and Patient Outcome in Prostate Cancer

Prognostic information was available for 12,457 (80.6%) of the 15,455 successfully analyzed prostate cancers. In these tumors, PSAP staining was absent in 3.4%, weak in 11.1%, moderate in 24.1%, and strong in 61.4% of cases. A low level of PSAP staining was significantly linked to a high Gleason score (*p* < 0.0001), advanced pT stage (*p* < 0.0001), nodal metastasis (*p* < 0.0001), high preoperative PSA values (*p* < 0.0001), and early PSA recurrence (Table 2 and Figure 3 *p* < 0.0001). Because a low level of PSAP immunostaining was strongly linked to TMPRSS2:ERG fusion, as detected by IHC and FISH (*p* < 0.0001 each; Appendix A), associations with tumor phenotype and prognosis were also separately analyzed in cohorts of ERG-negative and positive tumors (Appendix A). These analyses showed that the link between reduced PSAP expression and unfavorable tumor features was mainly driven by ERG-negative tumors. In addition, the association with PSA recurrence was stronger in ERG-negative (Figure 3; *p* < 0.0001) than in ERG-positive cancers (*p* = 0.0012; Figure 3). A low level of PSAP immunostaining was also associated with high androgen receptor levels in both ERG-negative (*p* < 0.0001) and ERG-positive cancers (*p* < 0.0001, Figure 4).

### 3.5. Multivariate Analysis

To evaluate the clinical relevance of PSAP expression, multivariate analyses modeling the different clinical scenarios were carried out. Scenario 1 included all postoperatively available parameters, including pT, pN, surgical margin status, preoperative PSA value, and Gleason grade obtained after a morphological evaluation of the entire resected prostate. Scenario 2 included all postoperatively available parameters with the exception of pN. Our approach was based on the fact that the indication and extent of lymph node dissection is not standardized regarding surgical therapy for prostate cancer. Case numbers can also be increased if the nodal stage is excluded from multivariate analyses. We also calculated an additional scenario to recreate the preoperative situation as closely as possible. The third scenario included preoperative PSA, clinical tumor stage (cT stage), and the Gleason grade obtained from the prostatectomy specimen. Because the definite Gleason grade obtained from the prostatectomy specimen is more precise than the Gleason grade from the pre-surgical biopsy (which is prone to sampling errors and consequent under-grading in more than one-third of cases [29]), we added a fourth scenario, in which the preoperative Gleason grade obtained on the original biopsy was combined with preoperative PSA and clinical (cT) tumor stage. In ERG-negative cancers, low PSAP proved to be an independent predictor of poor prognosis in scenarios 2, 3, and 4, while PSAP measurement did not provide independent prognostic information in ERG-positive cancers (Appendix A). 

## 4. Discussion

Our analysis identified PSAP positivity in 96.9% of 15,455 prostate cancers. This fits well with earlier data describing PSAP positivity in 59–95% of prostate cancers [6,7,8]. Altogether, these data identify PSAP immunohistochemistry as a highly sensitive marker for the identification of the prostatic origin of cancerous tissue, such as metastases of unknown origin. That only 4 of 13,611 extra-prostatic cancers showed PSAP immunostaining demonstrates that our PSAP assay exhibits a high (99.97%) level of specificity for prostate cancer. Among 127 surveyed extra-prostatic tumor entities, PSAP positivity was only observed in one of 129 (0.8%) gastric adenocarcinomas and in 3 of 94 (3.2%) pancreatic neuroendocrine tumors. Earlier studies have described PSAP expression in 0–11% of pancreatic neuroendocrine tumors [17,30], while two studies failed to find PSAP expression in gastric adenocarcinomas [30,31]. It is noteworthy that PSAP expression has earlier been described to occur at relevant frequencies in various other tumor entities, including colorectal cancers, NSCLC cancers, ovarian cancers, pancreatic cancers, breast cancers, renal cancers, and carcinoid tumors of the gastrointestinal tract (studies summarized in Appendix A). Although high numbers of several of these entities were analyzed in our study, we were unable to identify any PSAP-positive cases. That the fraction of IHC-positive tumors varies between studies reflects several inherent issues of immunohistochemistry (summarized in [32]). Published data on the immunohistochemically determined positivity rates are, thus, highly variable for most, if not all, proteins that have been analyzed in several different laboratories [33,34].

With respect to the very large numbers of samples included in our study, we carefully validated our immunohistochemical PSAP assay before the TMAs were stained. 

Our validation approach followed the recommendations of the International Working Group for Antibody Validation (IWGAV). It was proposed that antibody validation should include a comparison of two different independent antibodies or, alternatively, a comparison between IHC and the expression data obtained by another independent method [35]. We applied both approaches in this project. RNA data that were obtained in three independent RNA screening studies, including the Human Protein Atlas (HPA) RNA-seq tissue dataset [36], the FANTOM5 project [37], and the Genotype-Tissue Expression (GTEx) project [38] are particularly useful for the validation of immunostaining obtained from antibodies that are directed against highly tissue-specific proteins such as PSAP. These studies had identified PSAP RNA only in the prostate and, to a very small extent, in the kidney (https://www.proteinatlas.org/ENSG00000014257-ACP3/summary/rna (accessed on 7 July 2023)). We employed two different antibodies for the purpose of validating our IHC assay. That the IHC analysis of normal tissues resulted in the complete restriction of PSAP staining to these organs, and that the same cell types as detected by MSVA-452M were also positive by PASE/4LJ, strongly validates our assay. The use of a very broad range of 76 different normal tissues for antibody validation ensures a high likelihood of detecting undesired cross-reactivities because virtually all proteins occurring in the normal cells of adult humans were subjected to our validation experiment. 

Our data also show that the PSAP expression level in tumor cells is a strong prognostic feature in prostate cancer. It remains unclear why cancers with reduced PSAP expression show higher tumor aggressiveness. Functional in vitro and in vivo studies have found a relationship between reduced or absent PSAP expression and the elevated phosphorylation of ErbB-2 and PI3K, increased cell growth, increased tumorigenicity, and the development of prostatic intraepithelial neoplasia (PIN) and adenocarcinoma in situ (CIS) (summarized in [5]). Since PSAP production is an important function of normal glandular cells of the prostate, one might also speculate that a deficiency in PSAP production might represent a subtle sign of cellular dedifferentiation. In that case, PSAP loss would represent a differentiation marker, rather than indicating the tumor-protective role of PSAP in prostate epithelial cells.

The availability of molecular data derived from previous studies utilizing the same set of TMAs enabled us to assess the associations between PSAP expression and molecular features of particular interest. We chose *TMPRSS2:ERG* fusion because this is the most frequent molecular alteration found in prostate cancer, and IHC data on AR expression because of its known interaction with PSAP. Finding a strong association between decreased PSAP expression and high levels of the androgen receptor protein is consistent with earlier functional data. Henttu et al. showed decreased PSAP expression in androgen-treated LNCaP cells, suggesting the negative regulation of PSAP expression by AR [39]. In addition, it has been suggested that decreased PSAP expression in hormone-refractory prostate cancer cells, which express functional AR, leads to the hyperphosphorylation of HER-2 and the androgen-independent activation of AR-signaling (summarized in [40]). 

This *TMPRSS2:ERG* fusion affects approximately half of all prostate cancers and predominates in patients of a younger age. The fusion causes the expression of the transcription factor ERG [41,42], which governs the activity of more than 1600 genes in prostate epithelial cells [22,43]. Our data identify the PSAP protein as ERG-dependent, with higher expression levels in ERG-negative than in ERG-positive cancers. That the prognostic role of PSAP was particularly strong in ERG-negative but was less prominent in ERG-positive cancers is in line with various earlier studies describing prognostic molecular features in prostate cancer that were either restricted to ERG-positive [44,45,46] or ERG-negative cancers [47,48]. These observations could be explained by activating or mitigating the effects of ERG-regulated proteins on the biological effects of various molecular features that can impact cancer aggressiveness. This dependence of the prognostic impact of individual biomarkers on specific molecular tumor characteristics may pose a major challenge for the development of prognostic cancer tests that can be used for any prostate cancer patient.

As the prognostic information derived from PSAP measurement was independent of established prognostic features in ERG-negative cancers, our data suggest that the quantification of PSAP protein levels may provide clinically useful information for this group of patients. In the future, we expect that multiparameter tests combining the analysis of multiple different prognostic tumor features may deliver enough relevant prognostic information to leverage its routine use in the evaluation of newly diagnosed prostate cancers. The rapidly emerging field of multicolor immunohistochemistry, allowing the simultaneous tumor cell-specific measurement of up to 40 protein markers, may prove to be particularly useful for such measurements (summarized in [49]).

## 5. Conclusions

Our data show the high sensitivity and specificity of PSAP IHC for the distinction of prostate carcinoma from other tumor entities. The independent association of decreased PSAP expression with adverse outcomes in ERG-negative prostate cancer makes PSAP measurement a candidate marker for inclusion in multiparameter prognostic panels.

## Figures and Tables

**Figure 1 diagnostics-13-03242-f001:**
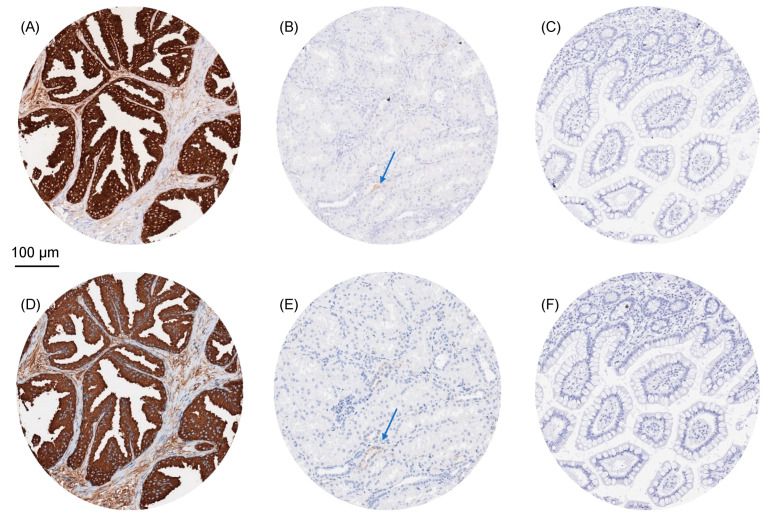
PSAP immunostaining of normal tissues. The panels show a comparison of the immunostaining obtained by two independent PSAP antibodies (MSVA-452M, PASE/4LJ). Using MSVA-452M, strong cytoplasmic PSAP positivity occurred in the acinar epithelial cells, while staining was weaker in some stroma cells of the prostate (**A**), and focal cytoplasmic staining (arrow) was seen in a few tubular cells of the kidney (**B**). PSAP staining was absent in the colon mucosa (**C**). Using clone PASE/4LJ, staining of the identical cell types was seen in the prostate (**D**) and the kidney (**E**), while the colon mucosa was also negative (**F**). Images (**A**–**F**) are taken from consecutive tissue sections.

**Figure 2 diagnostics-13-03242-f002:**
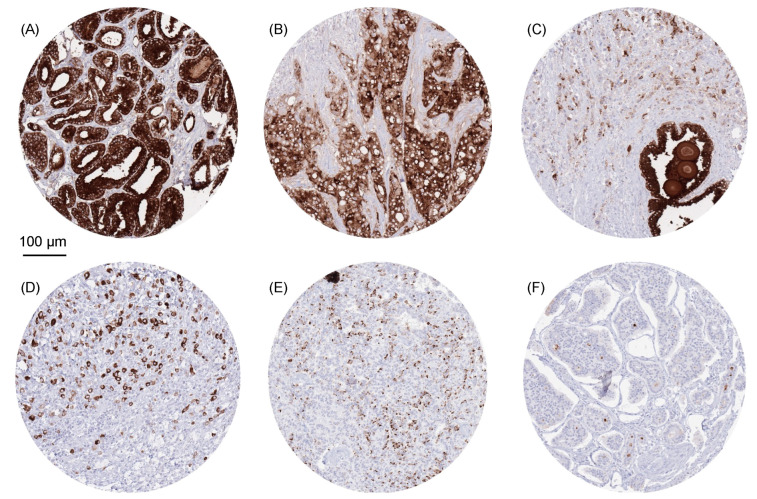
PSAP immunostaining in cancer. The panels show strong cytoplasmic PSAP positivity in Gleason 3 + 3 = 6 (**A**) and Gleason 5 + 5 = 10 (**B**) adenocarcinomas of the prostate, while PSAP staining is markedly decreased compared to a normal prostatic gland in another Gleason 5 + 5 = 10 cancer (**C**). Cytoplasmic PSAP staining of variable intensity is also seen in diffuse-type gastric adenocarcinoma (**D**) and two neuroendocrine tumors of the pancreas (**E**,**F**).

**Figure 3 diagnostics-13-03242-f003:**
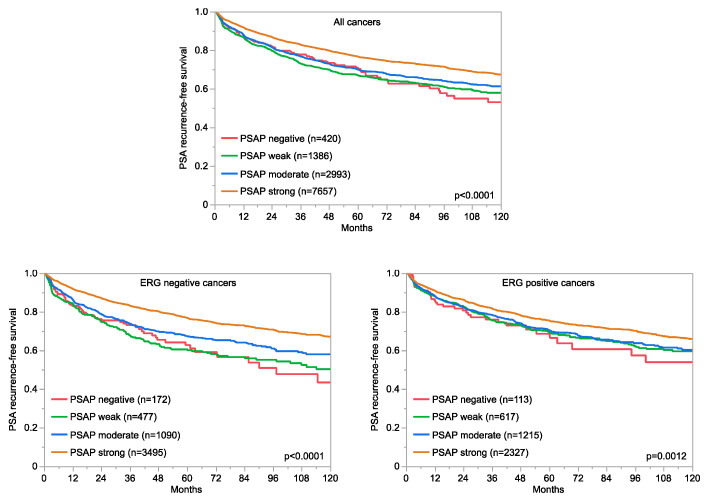
PSAP immunostaining and prognosis in prostate cancer.

**Figure 4 diagnostics-13-03242-f004:**
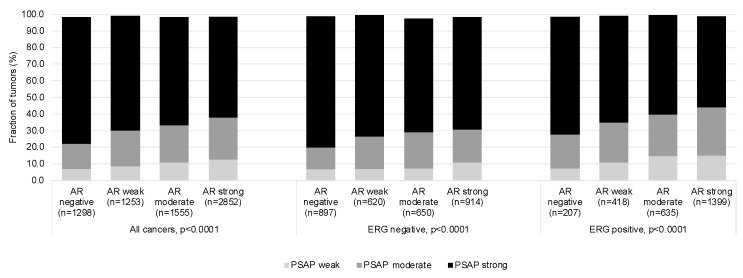
PSAP and androgen receptor immunostaining.

**Table 1 diagnostics-13-03242-t001:** PSAP immunostaining in human tumors.

			PSAP Immunostaining Result				
	Tumor Entity	on TMA (n)	Analyzable (n)	Negative (%)	Weak (%)	Moderate (%)	Strong (%)
Tumors of the skin	Pilomatricoma	35	33	100.0	0.0	0.0	0.0
	Basal cell carcinoma of the skin	88	55	100.0	0.0	0.0	0.0
	Benign nevus	29	28	100.0	0.0	0.0	0.0
	Squamous cell carcinoma of the skin	90	76	100.0	0.0	0.0	0.0
	Malignant melanoma	48	46	100.0	0.0	0.0	0.0
	Merkel cell carcinoma	46	41	100.0	0.0	0.0	0.0
Tumors of the head and neck	Squamous cell carcinoma of the larynx	110	96	100.0	0.0	0.0	0.0
	Squamous cell carcinoma of the pharynx	60	47	100.0	0.0	0.0	0.0
	Oral squamous cell carcinoma (floor of the mouth)	130	116	100.0	0.0	0.0	0.0
	Pleomorphic adenoma of the parotid gland	50	48	100.0	0.0	0.0	0.0
	Warthin tumor of the parotid gland	104	98	100.0	0.0	0.0	0.0
	Adenocarcinoma, NOS (papillary cystadenocarcinoma)	14	13	100.0	0.0	0.0	0.0
	Salivary duct carcinoma	15	11	100.0	0.0	0.0	0.0
	Acinic cell carcinoma of the salivary gland	181	140	100.0	0.0	0.0	0.0
	Adenocarcinoma NOS of the salivary gland	109	74	100.0	0.0	0.0	0.0
	Adenoid cystic carcinoma of the salivary gland	180	105	100.0	0.0	0.0	0.0
	Basal cell adenocarcinoma of the salivary gland	25	23	100.0	0.0	0.0	0.0
	Basal cell adenoma of the salivary gland	101	90	100.0	0.0	0.0	0.0
	Epithelial-myoepithelial carcinoma of the salivary gland	53	51	100.0	0.0	0.0	0.0
	Mucoepidermoid carcinoma of the salivary gland	343	257	100.0	0.0	0.0	0.0
	Myoepithelial carcinoma of the salivary gland	21	20	100.0	0.0	0.0	0.0
	Myoepithelioma of the salivary gland	11	11	100.0	0.0	0.0	0.0
	Oncocytic carcinoma of the salivary gland	12	12	100.0	0.0	0.0	0.0
	Polymorphous adenocarcinoma, low grade, of the salivary gland	41	34	100.0	0.0	0.0	0.0
	Pleomorphic adenoma of the salivary gland	53	35	100.0	0.0	0.0	0.0
Tumors of the lung, pleura, and thymus	Adenocarcinoma of the lung	246	167	100.0	0.0	0.0	0.0
	Squamous cell carcinoma of the lung	130	60	100.0	0.0	0.0	0.0
	Small cell carcinoma of the lung	20	16	100.0	0.0	0.0	0.0
	Mesothelioma, epithelioid	39	31	100.0	0.0	0.0	0.0
	Mesothelioma, biphasic	76	64	100.0	0.0	0.0	0.0
	Thymoma	29	29	100.0	0.0	0.0	0.0
	Lung, neuroendocrine tumor (NET)	19	17	100.0	0.0	0.0	0.0
Tumors of the female genital tract	Squamous cell carcinoma of the vagina	78	64	100.0	0.0	0.0	0.0
	Squamous cell carcinoma of the vulva	130	117	100.0	0.0	0.0	0.0
	Squamous cell carcinoma of the cervix	130	124	100.0	0.0	0.0	0.0
	Endometrioid endometrial carcinoma	236	225	100.0	0.0	0.0	0.0
	Endometrial serous carcinoma	82	72	100.0	0.0	0.0	0.0
	Carcinosarcoma of the uterus	48	40	100.0	0.0	0.0	0.0
	Endometrial carcinoma, high grade, G3	13	13	100.0	0.0	0.0	0.0
	Endometrial clear cell carcinoma	8	7	100.0	0.0	0.0	0.0
	Endometrioid carcinoma of the ovary	110	88	100.0	0.0	0.0	0.0
	Serous carcinoma of the ovary	559	451	100.0	0.0	0.0	0.0
	Mucinous carcinoma of the ovary	96	76	100.0	0.0	0.0	0.0
	Clear cell carcinoma of the ovary	50	38	100.0	0.0	0.0	0.0
	Carcinosarcoma of the ovary	47	38	100.0	0.0	0.0	0.0
	Brenner tumor	9	9	100.0	0.0	0.0	0.0
Tumors of the breast	Invasive breast carcinoma of no special type	126	69	100.0	0.0	0.0	0.0
	Lobular carcinoma of the breast	123	106	100.0	0.0	0.0	0.0
	Medullary carcinoma of the breast	15	15	100.0	0.0	0.0	0.0
	Tubular carcinoma of the breast	18	18	100.0	0.0	0.0	0.0
	Mucinous carcinoma of the breast	22	21	100.0	0.0	0.0	0.0
	Phyllodes tumor of the breast	50	50	100.0	0.0	0.0	0.0
Tumors of the digestive system	Adenomatous polyp, low-grade dysplasia	50	49	100.0	0.0	0.0	0.0
	Adenomatous polyp, high-grade dysplasia	50	49	100.0	0.0	0.0	0.0
	Adenocarcinoma of the colon	956	740	100.0	0.0	0.0	0.0
	Adenocarcinoma of the small intestine	10	6	100.0	0.0	0.0	0.0
	Gastric adenocarcinoma, diffuse type	226	129	99.2	0.0	0.0	0.8
	Gastric adenocarcinoma, intestinal type	224	130	100.0	0.0	0.0	0.0
	Gastric adenocarcinoma, mixed type	62	48	100.0	0.0	0.0	0.0
	Adenocarcinoma of the esophagus	133	65	100.0	0.0	0.0	0.0
	Squamous cell carcinoma of the esophagus	124	40	100.0	0.0	0.0	0.0
	Squamous cell carcinoma of the anal canal	91	74	100.0	0.0	0.0	0.0
	Cholangiocarcinoma	50	49	100.0	0.0	0.0	0.0
	Hepatocellular carcinoma	50	50	100.0	0.0	0.0	0.0
	Ductal adenocarcinoma of the pancreas	662	482	100.0	0.0	0.0	0.0
	Pancreatic/Ampullary adenocarcinoma	119	75	100.0	0.0	0.0	0.0
	Acinar cell carcinoma of the pancreas	15	14	100.0	0.0	0.0	0.0
	Gastrointestinal stromal tumor (GIST)	50	49	100.0	0.0	0.0	0.0
	Appendix, neuroendocrine tumor (NET)	22	12	100.0	0.0	0.0	0.0
	Colorectal, neuroendocrine tumor (NET)	11	9	100.0	0.0	0.0	0.0
	Ileum, neuroendocrine tumor (NET)	49	45	100.0	0.0	0.0	0.0
	Pancreas, neuroendocrine tumor (NET)	98	94	96.8	2.1	1.1	0.0
	Colorectal, neuroendocrine carcinoma (NEC)	12	8	100.0	0.0	0.0	0.0
	Gallbladder, neuroendocrine carcinoma (NEC)	4	4	100.0	0.0	0.0	0.0
	Pancreas, neuroendocrine carcinoma (NEC)	14	14	100.0	0.0	0.0	0.0
Tumors of the urinary system	Urothelial carcinoma, pT2-4 G3	1207	624	100.0	0.0	0.0	0.0
	Small cell neuroendocrine carcinoma of the bladder	18	17	100.0	0.0	0.0	0.0
	Sarcomatoid urothelial carcinoma	25	24	100.0	0.0	0.0	0.0
	Clear cell renal cell carcinoma	858	731	100.0	0.0	0.0	0.0
	Papillary renal cell carcinoma	255	197	100.0	0.0	0.0	0.0
	Clear cell (tubulo) papillary renal cell carcinoma	21	18	100.0	0.0	0.0	0.0
	Chromophobe renal cell carcinoma	131	90	100.0	0.0	0.0	0.0
	Oncocytoma of the kidney	177	117	100.0	0.0	0.0	0.0
Tumors of the male genital organs	Adenocarcinoma of the prostate, Gleason 3 + 3	83	77	0.0	2.6	15.6	81.8
	Adenocarcinoma of the prostate, Gleason 4 + 4	80	67	4.5	7.5	20.9	67.2
	Adenocarcinoma of the prostate, Gleason 5 + 5	85	78	9.0	20.5	25.6	44.9
	Adenocarcinoma of the prostate (recurrence)	261	210	6.2	21.0	23.8	49.0
	Small cell neuroendocrine carcinoma of the prostate	17	16	68.8	12.5	18.8	0.0
	Seminoma	621	427	100.0	0.0	0.0	0.0
	Embryonal carcinoma of the testis	50	34	100.0	0.0	0.0	0.0
	Yolk sac tumor	50	38	100.0	0.0	0.0	0.0
	Teratoma	50	45	100.0	0.0	0.0	0.0
	Squamous cell carcinoma of the penis	80	61	100.0	0.0	0.0	0.0
Tumors of endocrine organs	Adenoma of the thyroid gland	114	106	100.0	0.0	0.0	0.0
	Papillary thyroid carcinoma	392	339	100.0	0.0	0.0	0.0
	Follicular thyroid carcinoma	158	133	100.0	0.0	0.0	0.0
	Medullary thyroid carcinoma	107	96	100.0	0.0	0.0	0.0
	Anaplastic thyroid carcinoma	45	42	100.0	0.0	0.0	0.0
	Adrenal cortical adenoma	50	39	100.0	0.0	0.0	0.0
	Adrenal cortical carcinoma	26	25	100.0	0.0	0.0	0.0
	Pheochromocytoma	50	50	100.0	0.0	0.0	0.0
Tumors of hematopoetic and lymphoid tissues	Hodgkin lymphoma	103	69	100.0	0.0	0.0	0.0
	Non-Hodgkin lymphoma	62	55	100.0	0.0	0.0	0.0
	Small lymphocytic lymphoma, B-cell type (B-SLL/B-CLL)	50	22	100.0	0.0	0.0	0.0
	Diffuse large B cell lymphoma (DLBCL)	114	81	100.0	0.0	0.0	0.0
	Follicular lymphoma	88	51	100.0	0.0	0.0	0.0
	T-cell non-Hodgkin lymphoma	24	11	100.0	0.0	0.0	0.0
	Mantle cell lymphoma	18	7	100.0	0.0	0.0	0.0
	Marginal zone lymphoma	16	8	100.0	0.0	0.0	0.0
	Diffuse large B-cell lymphoma (DLBCL) in the testis	16	13	100.0	0.0	0.0	0.0
	Burkitt lymphoma	5	3	100.0	0.0	0.0	0.0
Tumors of soft tissue and bone	Tendosynovial giant cell tumor	45	43	100.0	0.0	0.0	0.0
	Granular cell tumor	53	43	100.0	0.0	0.0	0.0
	Leiomyoma	50	49	100.0	0.0	0.0	0.0
	Leiomyosarcoma	87	85	100.0	0.0	0.0	0.0
	Liposarcoma	132	114	100.0	0.0	0.0	0.0
	Malignant peripheral nerve sheath tumor (MPNST)	13	12	100.0	0.0	0.0	0.0
	Myofibrosarcoma	26	26	100.0	0.0	0.0	0.0
	Angiosarcoma	73	65	100.0	0.0	0.0	0.0
	Angiomyolipoma	91	90	100.0	0.0	0.0	0.0
	Dermatofibrosarcoma protuberans	21	18	100.0	0.0	0.0	0.0
	Ganglioneuroma	14	13	100.0	0.0	0.0	0.0
	Kaposi sarcoma	8	6	100.0	0.0	0.0	0.0
	Neurofibroma	117	89	100.0	0.0	0.0	0.0
	Sarcoma, not otherwise specified (NOS)	75	72	100.0	0.0	0.0	0.0
	Paraganglioma	41	35	100.0	0.0	0.0	0.0
	Ewing sarcoma	23	16	100.0	0.0	0.0	0.0
	Rhabdomyosarcoma	7	7	100.0	0.0	0.0	0.0
	Schwannoma	121	95	100.0	0.0	0.0	0.0
	Synovial sarcoma	12	11	100.0	0.0	0.0	0.0
	Osteosarcoma	43	38	100.0	0.0	0.0	0.0
	Chondrosarcoma	38	26	100.0	0.0	0.0	0.0

**Table 2 diagnostics-13-03242-t002:** PSAP immunostaining and the prostate cancer phenotype.

		PSAP Immunostaining Result				
	n Evaluable	Negative (%)	Weak (%)	Moderate (%)	Strong (%)	*p*-Value
All cancers	15,455	3.1	10.7	25.1	61.1	
Tumor stage						
pT2	9896	3.2	8.6	23.4	64.8	<0.0001
pT3a	3413	2.3	12	27.3	58.3	
pT3b-4	2081	3.9	18.4	29.7	48	
Gleason score						
≤3 + 3	2983	3.3	8.2	20.9	67.7	<0.0001
3 + 4	8159	2.7	9.9	24.9	62.6	
3 + 4 Tert.5	714	2.5	10.4	27.9	59.2	
4 + 3	1504	3.6	12.2	25.9	58.3	
4 + 3 Tert.5	1069	3	15.1	31.5	50.4	
≥4 + 4	877	5.5	19.3	30.6	44.7	
quantitative Gleason score						
3 + 4 ≤ 5%	2101	3.9	8.8	23.2	64.1	<0.0001
3 + 4 6–10%	2016	2.4	9.1	23.7	64.8	
3 + 4 11–20%	1760	2.1	9.4	26.2	62.3	
3 + 4 21–30%	904	2	12.7	24.1	61.2	
3 + 4 31–49%	754	1.6	12.2	26.7	59.5	
4 + 3 50–60%	592	2.5	10.4	27.9	59.2	
4 + 3 61–80%	551	2.5	11	24	62.5	
4 + 3 > 80%	136	4.2	13.2	26.7	55.9	
Lymph node metastasis						
N0	9228	2.9	10.8	26.4	59.9	<0.0001
N+	1146	4.3	18	30.4	47.4	
Preoperative PSA level (ng/mL)						
<4	1886	4.4	12.6	25.5	57.5	<0.0001
4–10	1120	3.5	15.1	27.1	54.3	
10–20	9108	2.9	9.6	24.5	63	
>20	3244	2.7	11.1	25.9	60.4	
Surgical margin						
negative	12,348	3.3	9.8	24.7	62.2	<0.0001
positive	3050	2.4	14.1	26.9	56.6	

## Data Availability

Raw data can be made available on reasonable request.

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
