# Peer review of "Diagnostic Role and Prognostic Impact of PSAP Immunohistochemistry: A Tissue Microarray Study on 31,358 Cancer Tissues"

_diagnostics, 2023, doi:10.3390/diagnostics13203242_

Round 1
Reviewer 1 Report
General Comments
The authors analyzed 14,137 samples from 127 different tumor types, 17,747 prostate cancers, and 76 different normal tissue types by immunohistochemistry in a tissue microarray format to assess specificity and the prognostic impact of prostate-specific acid phosphatase (PSAP) staining. Their results show PSAP is highly specific for prostate and a lower expression of PSAP is associated with worse prostate cancer outcomes. These results are not novel, however, stratified analyses of PSAP expression in ERG positive and negative tumors does add some new information to the scientific literature. The rigor in which the IHC experiments were performed appears appropriate, however, some specific issues related to the tissue processing and interpretation of staining results are discussed below.
Major Comments
1. High percentage of unanalyzable tissues
About 25% of the tumor samples in the multi-tumor TMA were uninterpretable while just over 10% of the prostate tumor samples could not be interpreted. Can the authors comment on this difference is so large and what effect this missing data might have on their results?
2. Pathologist scoring of staining intensity
The definitions of how staining intensity was scored does not include the number and types of readers for the TMA slides. Also, were any internal measures of slide scoring reliability performed?
3. Measures of Effect for PSAP associations with outcome
The results presented of associations of PSAP with prostate cancer outcomes only include statistical significance levels, no effect measures. Effect measures would be helpful in comparing PSAP associations between groups such as ERG positive and ERG negative patients. Can they be added to the results?
Minor Comments
1. Supplementary table 4 could use a more descriptive title
none
Author Response
General Comments
The authors analyzed 14,137 samples from 127 different tumor types, 17,747 prostate cancers, and 76 different normal tissue types by immunohistochemistry in a tissue microarray format to assess specificity and the prognostic impact of prostate-specific acid phosphatase (PSAP) staining. Their results show PSAP is highly specific for prostate and a lower expression of PSAP is associated with worse prostate cancer outcomes. These results are not novel, however, stratified analyses of PSAP expression in ERG positive and negative tumors does add some new information to the scientific literature. The rigor in which the IHC experiments were performed appears appropriate, however, some specific issues related to the tissue processing and interpretation of staining results are discussed below.
Major Comments
- High percentage of unanalyzable tissues
About 25% of the tumor samples in the multi-tumor TMA were uninterpretable while just over 10% of the prostate tumor samples could not be interpreted. Can the authors comment on this difference is so large and what effect this missing data might have on their results?
Reply: We have explained the higher rate of interpretable cases of the prostate cancer TMA in the Results section, page 3, lines 126-130.
- Pathologist scoring of staining intensity
The definitions of how staining intensity was scored does not include the number and types of readers for the TMA slides. Also, were any internal measures of slide scoring reliability performed?
Reply: In our experience, the involvement of multiple readers does not lead to significant improvement of clinic-pathological associations in a 4-tier system as the extreme groups negative and strong – which are eventually decisive for the statistical relationship – are not subject to significant interobserver variability. Most of all, striking associations with established clinical markers e.g. in prostate cancer can hardly be reversed by re-grouping of tumors from one category in the adjacent next category in our 4-tier system that was applied and validated in hundreds of TMA studies. We commented on this issue briefly in the Materials & Methods section, page 3, lines 100-103.
- Measures of Effect for PSAP associations with outcome
The results presented of associations of PSAP with prostate cancer outcomes only include statistical significance levels, no effect measures. Effect measures would be helpful in comparing PSAP associations between groups such as ERG positive and ERG negative patients. Can they be added to the results?
Reply: We now added risk ratios to Supplementary Table 4.
Minor Comments
- Supplementary table 4 could use a more descriptive title
Reply: We modified the title to better describe its content.
Reviewer 2 Report
The authors describe the application of IHC of Prostate-specific acid phosphatase (PSAP) as a method to differentiate the origin of cancer lesion. Although the idea is not novel, the scope of the work presented is very impressive. The conclusions from the work should be tuned down: the data supports that the assay can be used to differentiate prostate cancer from other, but it is not a good prognostic/predictive marker – specific staining pattern is statistically associated with, for example, cancer stage – but statistics are not definite. i.e. – if a patient has a weak staining, it does not really tell anything prognostic, it just gives an association, and a weak one at that. The conclusions must be tuned down.
Minor comments:
1. We are in the era of sequencing, and the methodology is becoming more and more available. Why would this approach be better than a molecular approach such as sequencing? A negative result in PSPA would just indicate that the lesion is not prostate cancer, sequencing might suggest an origin.
2. The are many antibodies that are commercially available for PSPA. Why did the authors choose just two? A comparison of multiple antibodies would make more sense.
3. The two antibodies shown in figure 1 have different staining of the stroma. A better analysis of the stroma staining should be done – is it a real staining? Does the stroma express RNA of phosphatase? Do other PSAP AB stain the stroma? Why only prostate cancer stroma stains for PSAP?
4. The RNA validation suggested in discussion is not validation. The authors should pick tumors that are strong, weak, and negative, extract RNA and compare them with staining. Showing data from literature / databases is not validation.
Author Response
The authors describe the application of IHC of Prostate-specific acid phosphatase (PSAP) as a method to differentiate the origin of cancer lesion. Although the idea is not novel, the scope of the work presented is very impressive. The conclusions from the work should be tuned down: the data supports that the assay can be used to differentiate prostate cancer from other, but it is not a good prognostic/predictive marker – specific staining pattern is statistically associated with, for example, cancer stage – but statistics are not definite. i.e. – if a patient has a weak staining, it does not really tell anything prognostic, it just gives an association, and a weak one at that. The conclusions must be tuned down.
Minor comments:
- We are in the era of sequencing, and the methodology is becoming more and more available. Why would this approach be better than a molecular approach such as sequencing? A negative result in PSPA would just indicate that the lesion is not prostate cancer, sequencing might suggest an origin.
Reply: We better described the area of application of PSAP IHC on page 11 lines 238-241.
- There are many antibodies that are commercially available for PSPA. Why did the authors choose just two? A comparison of multiple antibodies would make more sense.
Reply: We better explained that use of multiple antibodies only serves the purpose of assay validation and was not for comparison antibodies (page 12, lines 269-270).
- The two antibodies shown in figure 1 have different staining of the stroma. A better analysis of the stroma staining should be done – is it a real staining? Does the stroma express RNA of phosphatase? Do other PSAP AB stain the stroma? Why only prostate cancer stroma stains for PSAP?
Reply: We now explained that the stroma staining is due to a contamination artifact in the results section, page 3, lines 135-138.
- The RNA validation suggested in discussion is not validation. The authors should pick tumors that are strong, weak, and negative, extract RNA and compare them with staining. Showing data from literature / databases is not validation.
Reply: We have now commented on the usefulness of public data for the validation of tissue specifc antibodies such as PSAP on page 12, lines 263-267.
